# Latent Trajectory: A New Framework for Deep Actor-Critic Reinforcement Learning with Uncertainty Quantification

**Frank Shih**                                                                *shihf@mskcc.org*
*Department of Epidemiology and Biostatistics*
*Memorial Sloan Kettering Cancer Center*
*New York, NY 10017*

**Faming Liang**                                                             *fmliang@purdue.edu*
*Department of Statistics*
*Purdue University*
*West Lafayette, IN 47907*

**Reviewed on OpenReview:** *https://openreview.net/forum?id=8B74xdaRHa*

## Abstract

Uncertainty quantification in deep learning is challenging due to the complexity of deep neural networks. This challenge is particularly pronounced in deep reinforcement learning (RL), where agents interact with stochastic environments. In deep actor-critic RL, this challenge is further exacerbated due to the interdependence between the actor and critic updates. Existing uncertainty quantification methods for RL are predominantly developed within the Bayesian framework. While these methods estimate the uncertainty of the value function, their confidence intervals are often misleading, with the coverage rate frequently falling well below the nominal level. To address this issue, we introduce a novel deep RL framework that treats transition trajectories as latent variables. Leveraging this framework, we propose an adaptive Stochastic Gradient Markov Chain Monte Carlo algorithm to train deep actor-critic models, which naturally accounts for the interdependence between the actor and critic updates. We provide theoretical guarantees for the convergence of the proposed method and offer empirical evidence for its effectiveness in uncertainty quantification of the value function. The proposed latent trajectory framework is highly flexible, allowing for the integration of advanced RL strategies to further enhance deep actor-critic learning.

## 1 Introduction

Reinforcement learning (RL) tackles sequential decision-making problems by designing an agent that interacts with its environment to learn an optimal policy, aiming to maximize a value function. Accurately quantifying the uncertainty of the value function is crucial for ensuring reliable and robust RL applications. However, this task is particularly challenging in the context of deep RL due to two main factors: the complex architectures of deep neural networks (DNNs) and the adaptive nature of the RL process. Even in supervised learning, accurately quantifying the uncertainty for DNNs has proven to be difficult (see, e.g., Blundell et al., 2015, Lakshminarayanan et al., 2017, and Sun et al., 2021), and the dynamic, evolving process in RL exacerbates this challenge (Osband et al., 2016; Bellemare et al., 2017). A significant step toward addressing this challenge has been made in Shih & Liang (2024), where deep $Q$-networks are simulated from their posterior distribution under the Kalman Temporal Difference (KTD) framework (Geist & Pietquin, 2010; Shashua & Mannor, 2020), enabling accurate quantification of the uncertainty of the $Q$-value function throughout the RL process. However, this approach is difficult to extend to deep actor-critic RL, where the presence of an additional actor network adds complexities to uncertainty quantification.

To address this challenge, we introduce a novel deep RL framework which treats transition trajectories as latent variables. Leveraging this framework, we propose an adaptive Stochastic Gradient Markov Chain

Monte Carlo (SGMCMC) algorithm to train deep actor-critic networks, which simultaneously updates the actor network through a stochastic gradient descent (SGD) step and samples from the conditional distribution of the critic network — conditioned on the current actor network — via an SGMCMC step. Under mild conditions, we establish the convergence of this adaptive SGMCMC algorithm. Specifically, we show that the parameters of the actor network converge in probability to a fixed point, while the parameters of the critic network converge weakly to a target distribution, thereby enabling accurate uncertainty quantification for the associated value function.

In summary, this study has made three primary contributions:

- We introduce a novel latent trajectory framework for training deep actor-critic models that naturally accounts for the interdependence between the actor and critic updates, simplifying theoretical analysis.

- We provide theoretical guarantees for the convergence of the proposed deep actor-critic RL method — the latent trajectory framework coupled with an adaptive SGMCMC algorithm, which ensures effective training for the actor network while enabling proper uncertainty quantification for the critic network and, consequently, the value function.

- The proposed latent trajectory framework is highly flexible, allowing for the integration of advanced RL strategies to enhance deep actor-critic training and uncertainty quantification.

**Related Works**  Convergence analysis for deep actor-critic RL is inherently challenging due to the interdependence between the actor and critic updates. To the best of our knowledge, only limited progress has been made in this area; see, e.g., Wang et al. (2020), Cayci et al. (2022), and Tian et al. (2023). Both Wang et al. (2020) and Cayci et al. (2022) considered single-layer neural networks and employed a double-loop strategy. In the inner loop, the critic network performs sufficiently many updates to accurately estimate the value function, ensuring that the actor network operates with a reliable approximation of the true value function. This setup enables the actor-critic RL to be analyzed as a gradient method with approximation error. In contrast, Tian et al. (2023) applied a small-gain approach that accommodates multi-layer networks but is limited to finite state spaces. In this work, we establish convergence guarantees for deep actor-critic RL with multi-layer networks and general state spaces, which can be either discrete or continuous.

The proposed method provides accurate uncertainty quantification for the critic network. While Bayesian methods have been widely adopted for uncertainty quantification in machine learning, applying them rigorously to deep actor-critic RL presents significant challenges. For example, Osband et al. (2018) introduced uncertainty in deep RL through randomized priors within a maximum a posteriori (MAP) framework. However, the posterior consistency for such priors remains unestablished, and inappropriate priors can lead to severely biased inference. The deep actor-critic setting further complicates the uncertainty quantification issue due to the interdependence between the actor and critic updates. Other methods, such as those based on bootstrapping (Osband et al., 2016; Tasdighi et al., 2024; Peer et al., 2021; Kalweit & Boedecker, 2017), deep ensembles (Mai et al., 2022), Bayesian dropout (Moerland et al., 2017) Gaussian processes (Geist & Pietquin, 2010; Engel et al., 2003), quantile regression (Dabney et al., 2017), Kalman temporal difference (Shashua & Mannor, 2020), and distributional RL (Bellemare et al., 2017; Clements et al., 2019), have been proposed for uncertainty quantification in RL. However, their theoretical guarantees have not been established in the deep actor-critic setting.

In contrast, our method explicitly accounts for the evolving nature of online deep RL and ensures that uncertainty in the critic network is accurately quantified, despite challenges posed by the interdependence between the actor and critic updates.

## 2    Preliminaries on Actor-Critic Models

We consider discounted, finite horizon policy optimization problems. Let $\theta$ and $\psi$ denote the parameters of the actor and critic networks, respectively. Let $(s_1, a_1, s_2, a_2, \dots)$ be the transition trajectory generated by a stochastic policy $\pi_\theta$, where each action $a_t$ is sampled from the distribution $\pi_\theta(a_t|s_t)$ and $t$ indexes the

state transitions. Let $Q_{\pi_\theta}(s_t, a_t)$ denote the Q-value corresponding to the state-action pair $(s_t, a_t)$ under the policy $\pi_\theta$. At each time step $t$, the agent receives an immediate reward $r_t = r(s_t, a_t)$. Let $R_t = \sum_{\tau=t}^n \gamma^{\tau-t} r_\tau$ denote the truncated return up to horizon $n$, which is an unbiased estimator of $Q_{\pi_\theta}(s_t, a_t)$. Let $V_\psi$ be the critic network approximation to the value function $V_{\pi_\theta}$. For convenience, we denote a single transition of the state and action as $x = (s, a)$, the return estimate as $R$. In this paper, we focus on the advantage actor-critic algorithm (Sutton et al., 2000; Schulman et al., 2018) with the advantage function expressed as:

$$A_\psi(s_t, a_t) = R_t - V_\psi(s_t), \tag{1}$$

where $A_\psi$ indicates the dependence of the advantage function on the critic network $\psi$.

The policy gradient (Sutton & Barto, 2018) for the advantage actor-critic algorithm takes the form

$$g_\psi^{ac}(\theta) = \mathbb{E}_{\pi_\theta}\Big[\sum_{t=1}^n A_\psi(s_t, a_t)\nabla_\theta \log \pi_\theta(a_t|s_t)\Big], \tag{2}$$

where the expectation is taken with respect to on-policy trajectories generated by $\pi_\theta$. Note that different parameterization strategies can be used for the advantage function. For example, one may parameterize the $Q$-function, using temporal difference (TD) or Monte Carlo methods to estimate it (Schulman et al., 2018). The parameters $\theta$ and $\psi$ are then iteratively updated using SGD algorithms till convergence, at which a solution to the following equation is reached:

$$g_\psi^{ac}(\theta) = 0, \tag{3}$$

which characterizes a (local) optimum of the policy objective (not necessarily the global optimum). However, the convergence theory for such an iterative optimization algorithm is hard to establish except for special cases under restrictive assumptions, such as linear function approximation (Chen et al., 2023; Wu et al., 2022), greedy policies (Holzleitner et al., 2020), shallow neural network approximation (Wang et al., 2020; Cayci et al., 2022), or finite state space (Tian et al., 2023). In practice, various RL strategies have been proposed, such as Advantage Actor-Critic (A2C), Asynchronous Advantage Actor-Critic (A3C) (Mnih et al., 2016), Proximal Policy Optimization (PPO) (Schulman et al., 2017), Soft Actor-Critic (SAC) (Haarnoja et al., 2018), and Deep Deterministic Policy Gradient (DDPG)(Lillicrap et al., 2019), which employ different tuning techniques for policy gradients to enhance the convergence and stability of the training process.

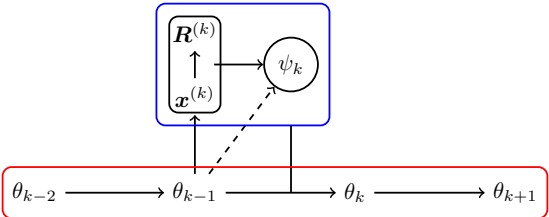

Figure 1: Latent Markov Sampling process, where $\psi_k$ is conditionally independent of $\theta_{k-1}$ given $\boldsymbol{x}^{(k)}$, i.e., $\psi_k \perp\!\!\!\perp \theta_{k-1}|\boldsymbol{x}^{(k)}$, and the dashed line indicates that including the latent variables $(\boldsymbol{x}^{(k)}, \boldsymbol{R}^{(k)})$ breaks the original dependence between $\psi_k$ and $\theta_{k-1}$.

Consider an actor-critic training process as depicted by Figure 1. Let $k$ index the updates of $\theta$ in the training process, and let $\boldsymbol{x}^{(k)} := \{x_t^{(k)}\}_{t=1}^n$ denote a batch of state-action transition trajectories drawn independently from the stationary distribution[1] $\pi(x|\theta_{k-1})$, where $x_t^{(k)} := (s_t^{(k)}, a_t^{(k)})$ and $n$ is the batch size. Additionally, let $\boldsymbol{R}^{(k)} = \{R_t^{(k)}\}_{t=1}^n$ be the estimated returns corresponding to $\boldsymbol{x}^{(k)}$. At each iteration $k$, a transition trajectory of size $n$ is generated from the policy $\pi_{\theta_{k-1}}$, the critic network parameter $\psi_k$ is then updated based on the transition trajectory.

---

[1]Given actor-network parameters $\theta_{k-1}$, the induced policy $\pi_{\theta_{k-1}}$ is fixed. Together with the environment transition dynamics, this policy induces a stationary distribution over state–action trajectories $\boldsymbol{x} = (s_0, a_0, s_1, a_1, \ldots)$, which we denote by $\pi(\boldsymbol{x} \mid \theta_{k-1})$ under standard ergodicity conditions.

The ultimate goal of RL is to learn an optimal policy, with the transition trajectory serving as an evolving path of the learning process and the critic network serving as an auxiliary guiding device for the path. Therefore, the transition trajectory $(\boldsymbol{x}^{(k)}, \boldsymbol{R}^{(k)})$ and the critic parameter $\psi_k$ can be naturally treated as latent variables that facilitate policy optimization. The whole RL process is Markovian as depicted by Figure 1. Leveraging the latent Markov sampling process, $\psi_k$ can also be viewed as a sample drawn from the conditional distribution $\pi(\psi_k|\theta_{k-1})$ by noting that

$$\pi(\psi_k|\theta_{k-1}) = \int \pi(\boldsymbol{x}^{(k)}, \boldsymbol{R}^{(k)}|\theta_{k-1})\pi(\psi_k|\boldsymbol{x}^{(k)}, \boldsymbol{R}^{(k)})d\boldsymbol{R}^{(k)}d\boldsymbol{x}^{(k)}. \tag{4}$$

To enhance the mathematical rigor of equation (4) in practical implementations, we assume that $\boldsymbol{x}^{(k)}$ is sampled from a pseudo-population[2] (denoted by $\Omega_{\theta_{k-1}}$) of size $\mathcal{N}$, while the pseudo-populations can vary for different values of $\theta_{k-1}$. Intuitively, when the pseudo-population is small, $\pi(x|\theta_{k-1})$ can be well approximated with a limited number of samples, thereby resulting in a good approximation to $\pi(\psi_k|\theta_{t-1})$ and strengthening the Markovian property of the process. The concept of pseudo-population allows for the flexibility of using different mini-batch size at different iterations, while ensuring the stability of the target conditional distribution $\pi(\psi|\theta)$. In what follows, we use $\pi_{\mathcal{N}}(\psi|\theta)$ to denote the conditional distribution of $\psi$ for a given value of $\theta$.

Furthermore, this latent Markov process view of RL allows us to frame the training of the actor network in terms of solving the following equation:

$$g(\theta) = \int g_{\psi}^{ac}(\theta)\pi_{\mathcal{N}}(\psi|\theta)d\psi = 0, \tag{5}$$

which accounts for the variation of $\psi$ in (3), and makes its solution (denoted by $\theta^*$) essentially invariant to the trajectories of $\psi$ and $(\boldsymbol{x}, \boldsymbol{R})$. Notably, $\theta^*$ is not necessarily unique. However, as with other applications of neural networks, this non-uniqueness of the optimal solution does not impact the network's performance. Equation (5) can be efficiently solved using an adaptive SGMCMC algorithm (Liang et al., 2022; Deng et al., 2019), leading to an innovative Latent Trajectory Framework (LTF) for training actor-critic models.

By examining equation (5) through the lens of Fisher's identity (see, e.g., Song et al., 2020), it becomes evident that $g(\theta)$ corresponds to the marginal gradient of the actor network when the critic network is treated as a latent variable. Consequently, training deep actor-critic networks falls into the class of problems amenable to stochastic approximation MCMC (see e.g., Benveniste et al., 1990). This formulation allows the interdependence between the actor and critic updates to be naturally handled by the proposed adaptive SGMCMC algorithm (see Section 3), leading to a simplified convergence analysis compared to the double-loop (Wang et al., 2020; Cayci et al., 2022) and small-gain (Tian et al., 2023) approaches.

## 3 A Latent Trajectory Framework for Actor-Critic Models

### 3.1 An Overview of the SGMCMC Algorithm

To solve equation (5), an adaptive SGMCMC algorithm can be applied. Each iteration of the algorithm consists of two steps:

1. ($\psi$-sampling) Simulate $\psi_k \sim \pi_{\mathcal{N}}(\psi|\theta_{k-1})$ by a SGMCMC algorithm.

2. ($\theta$-updating) Update $\theta_k = \theta_{k-1} + \upsilon_k\hat{g}_{\psi_k}^{ac}(\theta_{k-1})$, where $\upsilon_k$ denotes the step size used in the stochastic approximation procedure (Robbins & Monro, 1951), and $\hat{g}_{\psi_k}^{ac}(\theta_{k-1})$ represents an unbiased estimator of $g(\theta_{k-1})$.

---

[2]Given a fixed policy $\pi_\theta$ and a stationary, ergodic sequence of transition tuples, as the number $N$ of observed tuples tends to infinity, any consistent estimator $\hat{\psi}_N$ of the parameter $\psi$ in the value function $V_\psi$ collapses to a point (zero variance). To avoid the resulting degeneracy of the conditional distribution $\pi(\psi \mid \theta)$, we work with a pseudo-population so that $\pi(\psi \mid \theta)$ remains well defined and does not depend on $N$.

Under mild conditions, we establish the convergence of the proposed algorithm. Specifically, we show that $\|\theta_k - \theta^*\| \to 0$ in probability as $k \to \infty$, where $\theta^*$ denotes a solution to (5) as defined previously. Additionally, $\psi_k$ converges weakly (in 2-Wasserstein distance) to the conditional distribution $\pi_{\mathcal{N}}(\psi|\theta^*)$. Consequently, the algorithm enables proper uncertainty quantification for $\psi$-related quantities, such as the $V$- and $Q$-value functions, which are central to RL. Notably, uncertainty quantification for the value functions is generally beyond the capabilities of iterative optimization algorithms conventionally used to train deep actor-critic models (see, e.g., Tian et al. (2023)). Leveraging this latent trajectory formulation, we establish a actor-critic training framework that is essentially independent of the sample trajectory.

## 3.2 Adaptive Stochastic Gradient MCMC for Deep Actor-Critic Learning

To perform $\psi$-sampling using SGMCMC, we need to evaluate the gradient $\nabla_\psi \log \pi_{\mathcal{N}}(\psi_k|\theta_{k-1})$. This can be done using the following identity established in Song et al. (2020):

$$\nabla_\psi \log \pi(\psi|\theta) = \int \nabla_\psi \log \pi(\psi|\boldsymbol{z}, \theta) \pi(\boldsymbol{z}|\psi, \theta) d\boldsymbol{z},$$

where $\boldsymbol{z}$ denotes a latent variable. By treating the trajectory $(\boldsymbol{x}^{(k)}, \boldsymbol{R}^{(k)})$ as the latent variable, we can derive the following formula (refer to Appendix A for the derivation):

$$\nabla_\psi \log \pi_{\mathcal{N}}(\psi_k|\theta_{k-1}) = \int \nabla_\psi \log \pi_{\mathcal{N}}(\psi_k|\boldsymbol{x}^{(k)}, \boldsymbol{R}^{(k)}) \frac{\pi(\boldsymbol{R}^{(k)}|\boldsymbol{x}^{(k)}, \psi_k)}{\pi(\boldsymbol{R}^{(k)}|\boldsymbol{x}^{(k)})} \pi(\boldsymbol{x}^{(k)}, \boldsymbol{R}^{(k)}|\theta_{k-1}) d\boldsymbol{x}^{(k)} d\boldsymbol{R}^{(k)}, \qquad (6)$$

provided that the mini-batch size $n$ has been chosen to be sufficiently large, ensuring that $\boldsymbol{x}^{(k)}$ serves as a good representative of the underlying pseudo-population.

For convenience, though not a requirement, we assume that the reward distribution $\pi(R_t|x_t, \psi)$ is Gaussian, as given by

$$R_t|x_t, \psi \sim \mathcal{N}(V_\psi(s_t), \sigma^2). \qquad (7)$$

It is worth noting that the Gaussian assumption for the reward has also been employed under the Kalman Temporal Difference framework, see e.g., Geist & Pietquin (2010), Tripp & Shachter (2013), and Shashua & Mannor (2020).

**Remark 3.1** *How to evaluate $\nabla_\psi \log \pi_{\mathcal{N}}(\psi_k|\boldsymbol{x}^{(k)}, \boldsymbol{R}^{(k)})$? Based on (7), we have*

$$\begin{aligned}
\nabla_\psi \log \pi_{\mathcal{N}}(\psi_k|\boldsymbol{x}^{(k)}, \boldsymbol{R}^{(k)}) &= \nabla_\psi \log \pi_{\mathcal{N}}(\boldsymbol{R}^{(k)}|\boldsymbol{x}^{(k)}, \psi_k) + \nabla_\psi \log \pi(\psi_k) \\
&= \frac{\mathcal{N}}{n} \sum_{t=1}^n \nabla_\psi \log \pi(R_t^{(k)}|x_t^{(k)}, \psi_k) + \nabla_\psi \log \pi(\psi_k),
\end{aligned} \qquad (8)$$

*where $\pi(\psi_k)$ denotes the prior distribution of $\psi_k$.*

**Remark 3.2** *How to evaluate the importance weight $w_k = \frac{\pi(\boldsymbol{R}^{(k)}|\boldsymbol{x}^{(k)}, \psi_k)}{\pi(\boldsymbol{R}^{(k)}|\boldsymbol{x}^{(k)})}$? Since the numerator can be evaluated based on (7), we consider the evaluation of the denominator in this remark. One way is to evaluate the denominator based on the relationship:*

$$\pi(\boldsymbol{R}^{(k)}|\boldsymbol{x}^{(k)}) = \int \pi(\boldsymbol{R}^{(k)}|\boldsymbol{x}^{(k)}, \psi_k) \pi(\psi_k|\boldsymbol{x}^{(k)}) d\psi_k, \qquad (9)$$

*i.e., estimating the denominator by averaging the density $\pi(\boldsymbol{R}^{(k)}|\boldsymbol{x}^{(k)}, \psi_k)$ over a set of samples of $\psi_k$ drawn from $\pi(\psi_k|\boldsymbol{x}^{(k)})$. The auxiliary samples of $\psi_k$ can be simulated using a SGMCMC algorithm based on the following gradient estimation:*

$$\nabla_{\tilde{\psi}} \log \pi(\tilde{\psi}|\boldsymbol{x}^{(k)}) = \int \nabla_{\tilde{\psi}} \log \pi(\tilde{\psi}|\boldsymbol{x}^{(k)}, \tilde{\boldsymbol{R}}) \pi(\tilde{\boldsymbol{R}}|\boldsymbol{x}^{(k)}, \tilde{\psi}) d\tilde{\boldsymbol{R}}, \qquad (10)$$

*which can be estimated based on auxiliary samples of $\tilde{\boldsymbol{R}}$ drawn from $\pi(\tilde{\boldsymbol{R}}|\boldsymbol{x}^{(k)}, \tilde{\psi})$, as defined in (7).*

*Alternatively, one can estimate $\pi(\boldsymbol{R}^{(k)}|\boldsymbol{x}^{(k)})$ using the Nadaraya-Watson (NW) conditional density kernel estimator:*

$$\hat{\pi}(R|x) = \frac{\sum_{t=1}^{n} K_{h_2}(x - x_t^{(k)}) K_{h_1}(R - R_t^{(k)})}{\sum_{t=1}^{n} K_{h_2}(x - x_t^{(k)})}, \tag{11}$$

*where both $K_{h_1}(\cdot)$ and $K_{h_2}(\cdot)$ are Gaussian kernels, and $h_1$ and $h_2$ are their respective bandwidths. The NW estimator is known to be consistent provided $h_1 \to 0$, $h_2 \to 0$, and $nh_1h_2 \to \infty$ as $n \to \infty$ (Hyndman et al., 1996). Extensions of the NW estimator based on local polynomial smoothing are available, see e.g., Fan et al. (1996) and Gooijer & Zerom (2003). See Izbicki & Lee (2016) for an estimator in a high-dimensional regression setting.*

As a summary, we have Algorithm 1, which provides an efficient implementation for the proposed LTF. Although the algorithm is described to perform a single update of $\psi$ at each sampling step, multiple updates are also allowed. This does not interfere with the convergence theory of the algorithm.

---

**Algorithm 1** Latent Trajectory for A2C (LT-A2C)

---

1: **Step 0 (Initialization)**. Initialize actor network $\pi_{\theta_0}$ with learning rate sequence $\{v_k\}$, and initialize critic network $V_{\psi_0}$ with learning rate sequence $\{\epsilon_k\}$

2: **for** $k = 1, \ldots, \mathcal{K}$ **do**

3:    Generate trajectories $\boldsymbol{x}^{(k)} = \{x_t^{(k)}\}_{t=1}^{n}$ and returns $\boldsymbol{R}^{(k)} = \{R_t^{(k)}\}_{t=1}^{n}$ with policy $\pi_{\theta_{k-1}}$

4:    **Step 1 (Auxiliary sampling). Draw auxiliary $\tilde{\psi}$-samples.**

5:    **for** $j = 1, \ldots, m$ **do**

6:       **Presetting:** If $j = 1$, set $\tilde{\psi}_0 = \psi_{k-1}$

$$\tilde{\psi}_j = \tilde{\psi}_{j-1} + \frac{\delta_j}{2} \nabla_{\tilde{\psi}} \log \pi(\tilde{\psi}|\boldsymbol{x}^{(k)}) + \tilde{e}_j, \tag{12}$$

   where $\delta_j$ is the learning rate, $\nabla_{\tilde{\psi}} \log \pi(\tilde{\psi}|\boldsymbol{x}^{(k)}) = \frac{1}{L} \sum_{i=1}^{L} \nabla_{\tilde{\psi}} \log \pi(\tilde{\psi}|\boldsymbol{x}^{(k)}, \tilde{\boldsymbol{R}}_i)$ is calculated based on (10) using $L$ auxiliary samples of $\tilde{\boldsymbol{R}}$ drawn from $\pi(\tilde{\boldsymbol{R}}|\tilde{\psi}, \boldsymbol{x}^{(k)})$, and $\tilde{e}_j \sim N_p(0, \delta_j I_p)$.

7:    **end for**

8:    **Step 2 (Policy Evaluation). Sampling the critic network $\psi_k$.**

9:    **Computing importance weight:** calculate

$$\hat{w}_k = \frac{\pi(\boldsymbol{R}^{(k)}|\boldsymbol{x}^{(k)}, \psi_{k-1})}{\frac{1}{m} \sum_{j=1}^{m} \pi(\boldsymbol{R}^{(k)}|\boldsymbol{x}^{(k)}, \tilde{\psi}_j)}. \tag{13}$$

10:    **Sampling:** Draw $e_k \sim N_p(0, \frac{n}{\mathcal{N}} \epsilon_k I_p)$ and calculate

$$\psi_k = \psi_{k-1} + \frac{\epsilon_k}{2} \nabla_{\psi} \tilde{L}(\theta_{k-1}, \psi_{k-1}) + e_k,$$

   where the gradient term is given by

$$\nabla_{\psi} \tilde{L}(\theta_{k-1}, \psi_{k-1}) = \hat{w}_k \left\{ \sum_{t=1}^{n} \nabla_{\psi} \log \pi(R_t^{(k)}|x_t^{(k)}, \psi_{k-1}) + \frac{n}{\mathcal{N}} \nabla_{\psi} \log \pi(\psi_k) \right\}. \tag{14}$$

11:    **Step 3 (Policy Control). Updating the actor network $\theta_{k-1}$:** Compute advantage functions in (1) and update $\theta_{k-1}$ as

$$\theta_k = \theta_{k-1} + v_k \sum_{t=1}^{n} A_{\psi_k}(x_t^{(k)}, a_t^{(k)}) \nabla_\theta \log \pi_{\theta_{k-1}}(a_t^{(k)}|s_t^{(k)}).$$

12: **end for**

---

Furthermore, the proposed LTF is highly flexible and can be seamlessly integrated with various actor-critic learning strategies. Specifically, we can modify the critic network distribution $\pi_{\mathcal{N}}(\psi|\boldsymbol{R}_t, \boldsymbol{x}_t)$ and/or the policy gradient to align with different actor-critic learning strategies. For instance, we replaced the A2C policy gradient with the one used in PPO, resulting in the LT-PPO algorithm, as demonstrated in our numerical experiments. Potential alternatives for the critic network distribution will be discussed in the final section of the paper.

### 3.3 Convergence Theory

In the proposed LTF, we simulate $\psi_k \sim \pi(\psi_k|\theta_{k-1})$ with a SGMCMC algorithms while $\theta_k$ changes from iteration to iteration. We establish the $L_2$-convergence of $\{\theta_k : k = 1, 2, \ldots\}$ and the $\mathcal{W}_2$-convergence of $\{\psi_k : k = 1, 2, \ldots\}$ in Theorem 3.1 and Theorem 3.2, respectively. This implies that the actor network achieves an optimal policy, while the critic network converges weakly to the stationary distribution $\pi(\psi|\theta^*)$.

**Theorem 3.1 (Convergence of $\theta_k$)** *Suppose Assumptions A1-A5 (in Appendix B) hold, and the sample size of auxiliary $\tilde{\psi}$-samples is sufficiently large. Set the learning rate sequence $\{\epsilon_k\}_{k=1}^{\infty}$ and the step size sequence $\{v_k\}_{k=1}^{\infty}$ in the form:*

$$\epsilon_k = \frac{C_\epsilon}{c_\epsilon + k^\alpha}, \quad v_k = \frac{C_v}{c_v + k^\beta}, \tag{15}$$

*for some constants $C_\epsilon > 0$, $c_\epsilon > 0$, $C_v > 0$, $c_v > 0$, $\alpha, \beta \in (0, 1]$, and $\beta \leq \alpha \leq \min\{1, 2\beta\}$. Then there exists a root $\theta^* \in \{\theta : g(\theta) = 0\}$ such that*

$$\mathbb{E}\|\theta_k - \theta^*\|^2 \leq \xi v_k, \quad k \geq k_0, \tag{16}$$

*for some constants $k_0 > 0$ and $\xi = \lambda_0 + 6\sqrt{2}\varsigma_2(1 + G_\psi)^{1/2}G_\theta$, where $\lambda_0 > 0$ denotes a constant, $\varsigma_2$ is defined in Assumption A5, and $G_\psi$ and $G_\theta$ are given in Lemma B.2.*

The constants $G_\psi$ and $G_\theta$ depend, respectively, on the critic and actor networks, including their dimensions and structures. It is worth noting that condition (15) requires the learning rate of the critic network to decay no slower than that of the actor network, ensuring that valid samples of critic networks are used for the update of the actor network in the later period of the simulation. This aligns with conditions commonly seen in iterative optimization methods, which typically require an accurate estimate of the value function, see e.g., the double-loop methods (Wang et al., 2020; Cayci et al., 2022).

Let $\pi^* = \pi(\psi|\theta^*)$, let $T_k = \sum_{i=1}^{k} \epsilon_i$, and let $\mu_{T_k}$ denote the probability law of $\psi_k$. Theorem 3.2 establishes convergence of $\mu_{T_k}$ in 2-Wasserstein distance.

**Theorem 3.2 ($\mathcal{W}_2$-convergence of $\psi_k$)** *Suppose Assumptions A1-A6 (in Appendix B) hold, the sample size of auxiliary $\tilde{\psi}$-samples is sufficiently large, and the sequences $\{\epsilon_k\}_{k=1}^{\infty}$ and $\{v_k\}_{k=1}^{\infty}$ are set as in Theorem 3.1. Then, for any $k \in \mathbb{N}$,*

$$\mathcal{W}_2(\mu_{T_k}, \pi^*) \leq (\hat{C}_0 \delta_{\tilde{L}}^{1/4} + \tilde{C}_1 v_1^{1/4})T_k + \hat{C}_2 e^{-T_k/c_{LS}}, \tag{17}$$

*for some positive constants $\hat{C}_0$, $\hat{C}_1$, and $\hat{C}_2$, where $\mathcal{W}_2(\cdot, \cdot)$ denotes the 2-Wasserstein distance, $c_{LS}$ denotes the logarithmic Sobolev constant of $\pi^*$, and $\delta_{\tilde{L}}$ is a coefficient as defined in Assumption A3 and reflects the variation of the stochastic gradient $\nabla_\psi \tilde{L}(\theta_{k-1}, \psi_k)$.*

The right-hand side of (17) can be made sufficiently small by selecting a large enough mini-batch size, a sufficiently small value of $v_1$, and a large number of iterations.

We prove Theorems 3.1 and 3.2 by following the proofs of adaptive SGLD provided in Dong et al. (2023) and Liang et al. (2025), respectively (see Appendix B). The proof for the expression of $\xi$ can be found in Theorem A.1 of Dong et al. (2023). It is worth noting that the proposed LTF can also be implemented using an adaptive SGHMC algorithm (Liang et al., 2022). In this case, Theorems 3.1 and 3.2 can still be established similar to the convergence theory presented in Liang et al. (2022).

## 4 Experiments

In this section, we evaluate the performance and effectiveness of the proposed LTF in enhancing deep actor-critic RL. We conduct experiments in two environments: the Indoor Escape Environment, where we demonstrate the performance of the LTF-enhanced deep actor-critic methods in uncertainty quantification; and the PyBullet Environment (Ellenberger, 2018–2019), where we compare the performance of LTF-enhanced deep actor-critic methods to their vanilla counterparts on continuous control benchmarks, demonstrating the flexibility of the LTF in integrating different RL strategies to enhance deep actor-critic training. These experiments highlight the improvements in deep actor-critic RL achieved through the adoption of the LTF.

## 4.1 Indoor Escape Environment

Figure 2 depicts a simple indoor escape environment (Shih & Liang, 2024), where the state space consists of 100 grids and the agent's objective is to navigate to the goal positioned at the top right corner. The agent starts its task from the bottom left grid at time $t = 0$. For every time step $t$, the agent identifies its current position, represented by the coordinate $s \in \{(i, j) : i, j = 1, \ldots, 10\}$. Given a policy $\pi_\theta$, the agent chooses an action $a \in \{N, E, S, W\}$ with respect to the probability $\pi_\theta(a|s)$. The action taken by the agent determines the adjacent grid to which it moves. Following each action, the agent is awarded an immediate reward, $r_t$, drawn independently from the Gaussian distribution $\mathcal{N}(-1, 0.01)$.

We evaluate the performance of the proposed method from three aspects: (i) Policy Diversity: The policy, coded by the actor network and denoted as $\pi_\theta(a|s)$, should converge to a distribution that assigns equal probabilities to optimal actions and zero probability to others. (ii) Value Accuracy: The critic network is expected to accurately approximate the state value function $V^*(s)$ across the entire state space. (iii) Value Uncertainty: Algorithms should be capable of quantifying the uncertainty associated with the value function.

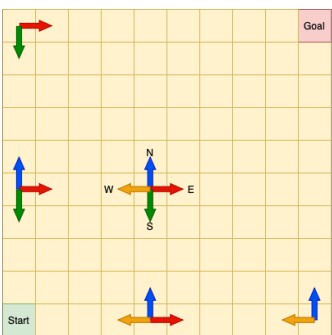

Figure 2: Indoor escape environment

To quantify policy diversity, we define the optimal policy distribution $\pi^*(\cdot|s)$ to be a probability distribution over all actions at state $s$, which is uniform over optimal actions and zero on sub-optimal actions. For a given policy $\pi_\theta(\cdot|s)$, the Kullback-Leibler (KL) divergence between $\pi^*$ and $\pi_\theta$, denoted by $D_{KL}(\pi^*\|\pi_\theta)$, can be used to measure the diversity of the policy distribution. It's worth noting that for most states, actions N and E are identically optimal. Hence, the policy $\pi_\theta(a|s)$ should assign equal probabilities on these two actions. Figure 3 visualizes the policy distribution $\pi_\theta$ at each state $s$. The left plot shows that the LTF achieves a nearly optimal policy distribution at each state $s$, which has a small value of $D_{KL}(\pi^*\|\pi_\theta)$. The right plot shows the policy distributions achieved by A2C, which is severely biased toward a single policy and has a large value of $D_{KL}(\pi^*\|\pi_\theta)$. Refer to Appendix C for the experimental settings.

Suppose that the actor network converges to a fixed policy $\pi_{\theta^*}$, and the state value function $V_\psi(s)$ coded by the critic network should be distributed around the optimal value function $V^*(s)$. To evaluate such estimation, we collect the last 1000 critic parameter updates to form a pool of $\psi$-samples, denoted by $\boldsymbol{\psi}_s = \{\hat{\psi}_i\}$, which naturally induces a sample pool of values $\mathbf{V}_s = \{V_\psi(\cdot)|\psi \in \boldsymbol{\psi}_s\}$. From the value sample pool, we can obtain a point estimate of the state value at state $s$ by calculating the sample average $\hat{V}(s) = \frac{1}{n}\sum_{i=1}^n V_{\hat{\psi}_i}(s, a)$. For inter-

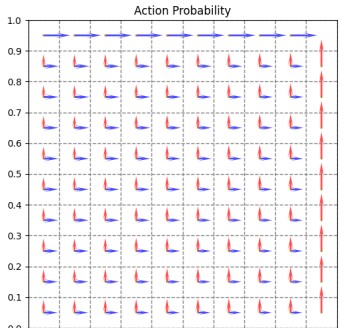
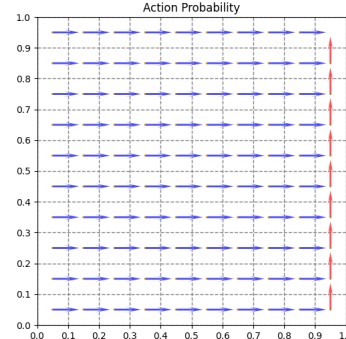

Figure 3: (Left) Policy distribution at each state $s$, achieved by LTF in a single run; (right) Policy distribution at each state $s$, achieved by A2C in a single run.

val estimation, we can achieve one-step value tracking by constructing a 95% prediction interval with the state value samples. We replicate each experiment 100 times and calculate the following three metrics: (i) the KL-divergence between $\pi^*$ and $\pi_{\theta^*}$, denoted by $D_{KL}(\pi^*\|\pi_{\theta^*})$, (ii) the mean squared error (MSE) between $\hat{V}(s)$ and $V^*(s)$, defined by $\text{MSE}(\hat{V}) = \mathbb{E}_{s\in\mathcal{S}}(\hat{V}(s) - V^*(s))^2$, where $\mathbb{E}(\cdot)$ denotes the empirical average over the state space $\mathcal{S}$; and (iii) the coverage rate (CR) of the 95% prediction intervals.

Table 1 demonstrates that, when enhanced with LTF, both A2C and PPO show substantial improvements across all three performance metrics. Compared to the vanilla A2C and PPO methods, which exhibit notable bias in value function estimation, the LTF-enhanced methods yield more consistent estimates and offer

Table 1: Comparison of A2C, PPO, LT-A2C, and LT-PPO for Indoor Escape. The number in the parentheses represents the standard deviation of the corresponding metric, which was calculated by averaging over 100 independent runs for each method.

| Algorithm | $\mathcal{N}$ | $D_{KL}(\pi^*\|\pi_{\theta^*})$ | MSE($\hat{V}$) | Coverage Rate | CI-Width |
|---|---|---|---|---|---|
| A2C | - | 4.647 (0.0729) | 0.53527 (0.03974) | 0.489 (0.0061) | 0.413 (0.0023) |
| LT-A2C | 10000 | 0.010 (0.0010) | 0.00038 (0.00001) | 0.947 (0.0004) | 0.457 (0.0009) |
| LT-A2C | 20000 | 0.014 (0.0014) | 0.00039 (0.00001) | 0.947 (0.0004) | 0.452 (0.0010) |
| LT-A2C | 40000 | 0.014 (0.0013) | 0.00033 (0.00001) | 0.947 (0.0004) | 0.449 (0.0009) |
| PPO | - | 4.773 (0.0893) | 0.56112 (0.04272) | 0.487 (0.0066) | 0.416 (0.0024) |
| LT-PPO | 10000 | 0.011 (0.0010) | 0.00041 (0.00001) | 0.947 (0.0004) | 0.458 (0.0009) |
| LT-PPO | 20000 | 0.009 (0.0009) | 0.00038 (0.00001) | 0.947 (0.0005) | 0.452 (0.0009) |
| LT-PPO | 40000 | 0.011 (0.0011) | 0.00032 (0.00001) | 0.947 (0.0004) | 0.449 (0.0008) |

reliable uncertainty quantification. Across all three performance metrics, LT-A2C and LT-PPO consistently outperform the original A2C and PPO methods, yielding lower MSE values, reduced KL-divergence, and improved coverage rates. The lower KL-divergence values suggest that the policy distribution more effectively converges to a uniform distribution over optimal actions, thereby enhancing exploration efficiency across the state space. These findings can be further visualized in Figure A1 and Figure A2. The former summarizes the results on KL-divergence and CI-Width, and the latter summarizes the results on MSE($\hat{V}$) and coverage rates. For MSE($\hat{V}$), the LTF-enhanced methods exhibit significantly smaller values and tighter box plots, reflecting greater training stability. In terms of uncertainty quantification, only the LTF-enhanced methods achieve coverage rates close to the nominal 95%. Furthermore, as the pseudo-population size increases, uncertainty quantification for the value function becomes more accurate, as indicated by reduced MSE values and narrower confidence interval widths.

Many existing methods claim the ability to quantify the uncertainty of the critic network. These methods often rely on approaches such as Bootstrap DQN, Quantile Regression DQN (distributional RL), and Random Prior Networks (RPN). While these methods provide estimates of uncertainty, they fail to construct statistically honest confidence intervals for the Q-function. As demonstrated in Table 2, we evaluated these methods in the Indoor Escape environment using a single neural network to approximate the $Q$-value function. None of them was able to accurately construct the 95% confidence interval for the $Q$-value, highlighting their limitations in uncertainty quantification: when neural networks are involved, the existing methods can be challenging to calibrate, often leading to inaccurate coverage rates of $Q$-values. This raises significant concerns about their ability in quantifying uncertainty for more complex actor-critic models.

Table 2: Comparison of different methods for Indoor Escape: (i) BootDQN: Bootstrapped DQN (Osband et al., 2016), (ii) Quantile regression (QR)-DQN: Distributional RL (Bellemare et al., 2017), (iii) RPN: Randomized Prior Networks (Osband et al., 2018). The number in the parentheses represents the standard deviation of the corresponding metric, which was calculated by averaging over 100 independent runs.

| Algorithm | MSE($\hat{Q}$) | Coverage Rate | CI-Width |
|---|---|---|---|
| BootDQN | 0.09979 (0.01609) | 0.388 (0.0186) | 0.188 (0.0032) |
| QR-DQN | 0.00459 (0.00028) | 0.821 (0.0089) | 0.278 (0.0033) |
| RPN: prior scale=0.1 | 0.03339 (0.00290) | 0.802 (0.0147) | 0.679 (0.0243) |
| RPN: prior scale=1.0 | 0.03724 (0.00412) | 0.816 (0.0157) | 0.693 (0.0243) |
| RPN: prior scale=5.0 | 0.03658 (0.00297) | 0.793 (0.0190) | 0.782 (0.0341) |

In the Indoor Escape environment, each episode is capped at 200 steps in our implementations. As a result, reaching the goal – without relying on additional exploration techniques like $\epsilon$-greedy – depends heavily on the ability of the method in performing parameter space exploration. Figure 4(a) plots the best-so-far reward versus time steps for each method: LT-A2C reaches the global optimum ($-18$) in every run, whereas A2C fails to do so in a substantial fraction of runs. Figure 4(b) reports the proportion of experiments in which each method successfully attained the optimal policy: LT-A2C reached the optimal policy in 100% of the runs, whereas A2C succeeded in only 78% of the runs. The comparison highlights the capability of the

LTF in parameter space exploration and suggests that it substantially outperforms A2C in identifying the optimal policy, while accurately quantifying the uncertainty of the resulting values.

Regarding computational complexity, we note that although LTF requires additional SGMCMC sampling for the critic network at each iteration, the per-iteration complexity of SGLD and SGHMC is comparable to that of SGD. As a result, the overall time complexity remains on par with standard iterative optimization methods, supporting the scalability of the proposed method. The LTF methods can be readily applied to large-scale neural networks.

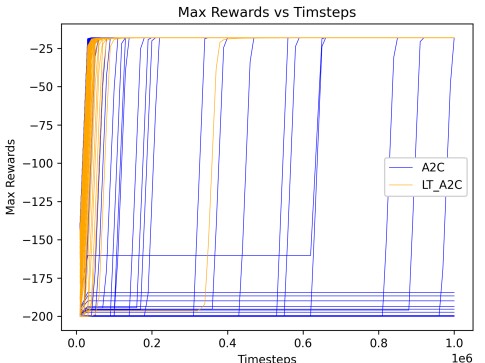 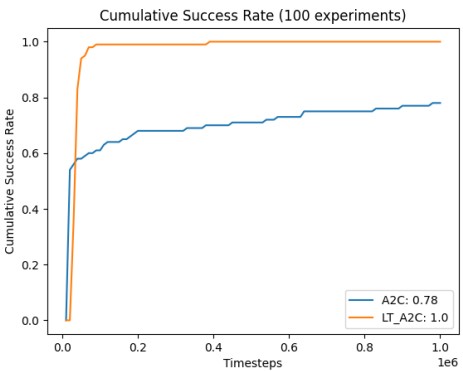

(a) Maximum reward achieved along time steps    (b) Success rate of achieving optimal policy

Figure 4: Optimal policy exploration for the Indoor Escape environment

## 4.2 PyBullet Environment

To demonstrate the applicability of the LTF-enhanced actor-critic RL methods in complex environments, we evaluate their performance on continuous control benchmarks using the RL Baselines3 Zoo (Raffin, 2020) and the PyBullet environment (Ellenberger, 2018–2019). We compare the LT-A2C and LT-PPO methods with their vanilla counterparts (A2C and PPO, respectively) across four continuous control tasks: HalfCheetah, Hopper, Reacher, and Walker. The hyperparameters for A2C and PPO follow the default configurations from RL Baselines3 Zoo (Raffin, 2020). Additional experimental details are provided in Appendix C.

Figure 5 presents the best reward up to the current iteration obtained by these methods for the control tasks. Each curve in the plots represents the mean of the best-so-far reward obtained by the corresponding method in 100 independent runs. The shaded region indicates the 95% confidence interval of the expected return, computed as the mean $\pm 1.96$ times the standard error. The comparison shows that both LT-A2C and LT-PPO outperform their vanilla counterparts in Walker-2D, while performing comparably in other control tasks. This underscores the flexibility of the proposed LTF, which allows for the integration of various advanced RL strategies to enhance actor-critic training while ensuring proper uncertainty quantification for the critic network and the resulting value function. Specifically, for PPO, its gradient clipping mechanism prevents the actor's policy from changing too drastically, helping ensure stable updates and improving the training of the actor-critic networks, even under the proposed LTF.

In terms of computation, the LT framework introduces an inner loop that samples the critic network (Algorithm 1), which increases training cost. However, the number of sampling steps need not be large; 5–10 steps typically suffice. To assess how this hyperparameter affects wall-clock time and performance, we conducted an ablation over the number of sampling steps, with results presented in Figure 6. The comparison shows that reward improvements slow after about 10 steps. Accordingly, we recommend using 10 sampling steps to reliably achieve good performance while maintaining efficient training.

A close inspection of Figure 6 shows that LT-A2C's performance continues to improve as the number of critic-network sampling steps increases. This, in turn, suggests that a high-quality SGMCMC algorithm could further enhance the method's performance, though it is not strictly essential.

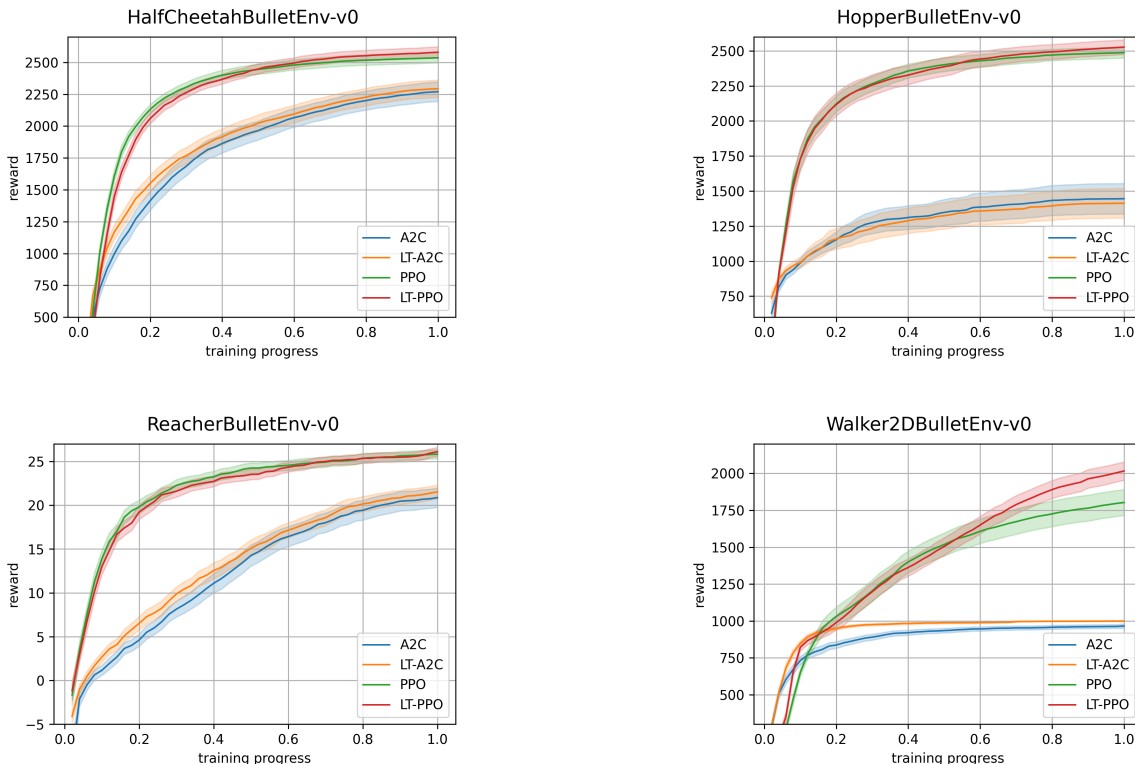

Figure 5: Comparison of A2C, PPO, LT-A2C, and LT-PPO for the PyBullet environments: training, evaluation/testing, and best/highest rewards (up to the current iteration) achieved during training. The results were summarized from 100 independent runs for each method.

## 5 Conclusion

In this paper, we have introduced a novel Latent Trajectory Framework, coupled with an adaptive SGMCMC algorithm, for training deep actor-critic networks. The proposed method naturally captures the interdependence between the actor and critic updates. We provide theoretical guarantees that, under mild conditions, the proposed method ensures consistency in parameter estimation for the actor network and weak convergence in parameter sampling for the critic network. Compared to existing iterative optimization methods, the proposed method ensures accurate uncertainty quantification for the critic network and, consequently, the value function, while offering greater flexibility and robustness in training the actor network. Existing Bayesian methods generally lack the mathematical rigor required to achieve this level of uncertainty quantification for deep actor-critic networks. Notably, the proposed method accomplishes these improvements without increasing the order of computational complexity compared to existing iterative optimization methods, making it scalable to large-scale DNNs.

In searching for high-reward policies, LTF is expected to outperform the vanilla baselines when the landscape of the actor-network objective is rugged. The added randomness from sampling critic networks – compared with iterative optimization – facilitates escape from local optima and speeds convergence to higher-reward policies. On smoother landscapes, LTF and iterative optimization-based methods perform similarly, with no loss from sampling.

The proposed LTF is also highly flexible. In addition to replacing the SGLD steps with SGHMC (Chen et al., 2014) or substituting the SGD step with Adam (Kingma & Ba, 2014), one can modify the policy control step and/or the critic network distribution to incorporate advanced RL strategies, further accelerating its convergence toward high-reward polices. In this paper, we have demonstrated that substituting the policy control step with a PPO update significantly enhances the training of actor-critic models in complex

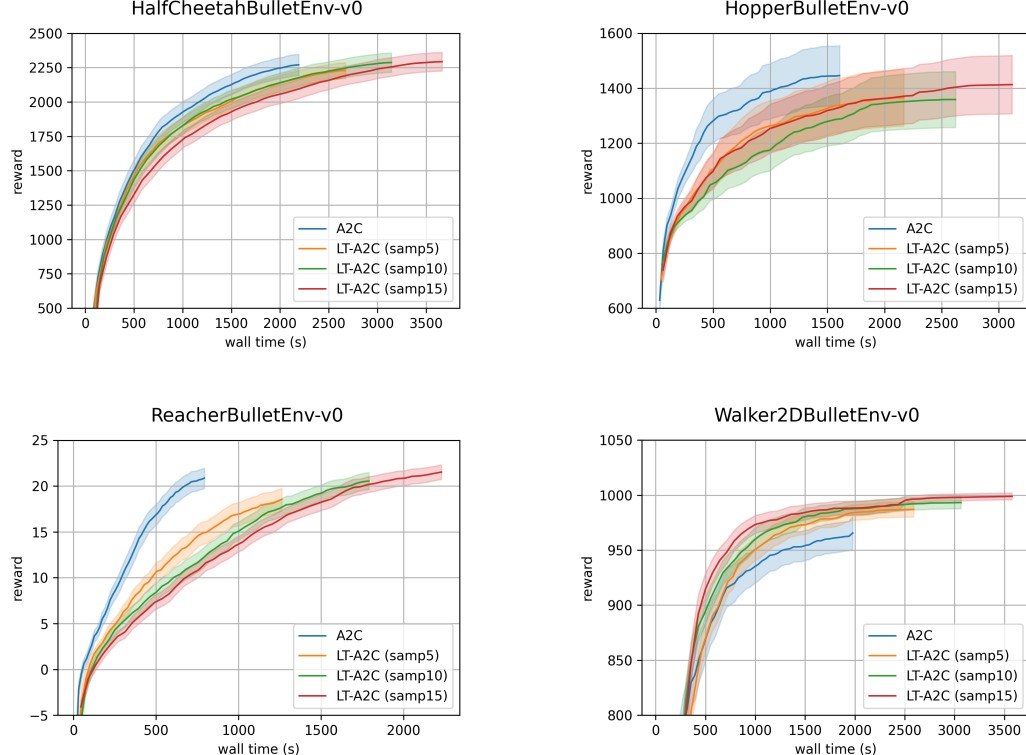

Figure 6: Comparison of A2C and LT-A2C (with varying numbers of sampling steps) on PyBullet environments. The plots show the best-so-far rewards (up to the current wall-clock time) during training. Results are averaged over 100 independent runs for each method.

environments. Similarly, the critic network distribution can be replaced with one derived from the critic update of an advanced RL strategy. For example, by leveraging the randomized ensembled double Q-learning (REDQ) strategy (Chen et al., 2021), we can replace the critic network distribution $\pi_{\mathcal{N}}(\psi|\boldsymbol{R}_t, \boldsymbol{x}_t)$ used in (8) with the following ensembled version:

$$\pi_{\mathcal{N}}(\boldsymbol{\Psi}|\boldsymbol{r}_t, \boldsymbol{x}_t) \propto \exp\left\{-\frac{\mathcal{N}}{|\boldsymbol{B}|}\sum_{\psi_i \in \boldsymbol{\Psi}}\sum_{(s.a.r,s') \in \boldsymbol{B}}(Q_{\psi_i}(s,a) - y)^2\right\}\pi(\boldsymbol{\Psi}),$$

where $\boldsymbol{\Psi} = \{\psi_1, \psi_2, \ldots, \psi_N\}$ denotes an ensemble of critic networks, $\boldsymbol{B}$ represents a mini-batch with size $|\boldsymbol{B}|$, and $y$ is defined as in Algorithm 1 of Chen et al. (2021). The detailed definition of $y$ is omitted here for simplicity. Based on other strategies, such as SUNRISE (Lee et al., 2021) and Optimistic Actor-Critic (OAC) (Ciosek et al., 2019), the critic network distribution can also be defined accordingly. In summary, the proposed LTF is able to accelerate the search for optimal policies by leveraging the strengths of modern RL strategies as well as the inherent randomness from sampling critic networks, while providing calibrated uncertainty quantification for the critic network and value functions. It is readily applicable to diverse actor–critic training tasks.

## Acknowledgments

Liang's research is supported in part by the NSF grant DMS-2210819 and the NIH grant R01-GM152717. Shih's research is partially supported by MSK Cancer Center Support Grant/Core Grant (P30 CA008748).

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

# Appendix

## A Derivation of Equation (6)

$$
\begin{aligned}
&\nabla_\psi \log \pi_\mathcal{N}(\psi_k | \theta_{k-1}) \\
&= \int \nabla_\psi \log \pi_\mathcal{N}(\psi_k | \boldsymbol{x}^{(k)}, \boldsymbol{R}^{(k)}, \theta_{k-1}) \pi(\boldsymbol{x}^{(k)}, \boldsymbol{R}^{(k)} | \psi_k, \theta_{k-1}) d\boldsymbol{x}^{(k)} d\boldsymbol{R}^{(k)} \\
&= \int \nabla_\psi \log \pi_\mathcal{N}(\psi_k | \boldsymbol{x}^{(k)}, \boldsymbol{R}^{(k)}) \frac{\pi(\boldsymbol{x}^{(k)}, \boldsymbol{R}^{(k)} | \psi_k, \theta_{k-1})}{\pi(\boldsymbol{x}^{(k)}, \boldsymbol{R}^{(k)} | \theta_{k-1})} \pi(\boldsymbol{x}^{(k)}, \boldsymbol{R}^{(k)} | \theta_{k-1}) d\boldsymbol{x}^{(k)} d\boldsymbol{R}^{(k)} \\
&= \int \nabla_\psi \log \pi_\mathcal{N}(\psi_k | \boldsymbol{x}^{(k)}, \boldsymbol{R}^{(k)}) \frac{\pi(\boldsymbol{R}^{(k)} | \boldsymbol{x}^{(k)}, \psi_k, \theta_{k-1}) \pi(\boldsymbol{x}^{(k)} | \psi_k, \theta_{k-1})}{\pi(\boldsymbol{R}^{(k)} | \boldsymbol{x}^{(k)}, \theta_{k-1}) \pi(\boldsymbol{x}^{(k)} | \theta_{k-1})} \pi(\boldsymbol{x}^{(k)}, \boldsymbol{R}^{(k)} | \theta_{k-1}) d\boldsymbol{x}^{(k)} d\boldsymbol{R}^{(k)} \\
&= \int \nabla_\psi \log \pi_\mathcal{N}(\psi_k | \boldsymbol{x}^{(k)}, \boldsymbol{R}^{(k)}) \frac{\pi(\boldsymbol{R}^{(k)} | \boldsymbol{x}^{(k)}, \psi_k)}{\pi(\boldsymbol{R}^{(k)} | \boldsymbol{x}^{(k)})} \pi(\boldsymbol{x}^{(k)}, \boldsymbol{R}^{(k)} | \theta_{k-1}) d\boldsymbol{x}^{(k)} d\boldsymbol{R}^{(k)},
\end{aligned} \tag{A1}
$$

provided that the mini-batch size $n$ has been chosen to be sufficiently large, ensuring that $\boldsymbol{x}^{(k)}$ serves as a good representative of the underlying pseudo-population. This guarantees that $\psi_k$ is conditionally independent of $\theta_{k-1}$ given $\boldsymbol{x}^{(k)}$, see Figure 1 for the graphical illustration of the latent Markov sampling process. Consequently, we have $\nabla_\psi \log \pi_\mathcal{N}(\psi_k | \boldsymbol{x}^{(k)}, \boldsymbol{R}^{(k)}, \theta_{k-1}) = \nabla_\psi \log \pi_\mathcal{N}(\psi_k | \boldsymbol{x}^{(k)}, \boldsymbol{R}^{(k)})$ and $\pi(\psi_k | \boldsymbol{x}^{(k)}) = \pi(\psi_k | \theta_{k-1})$. The former leads to the second equality in (A1), while the latter leads to the last equality in (A1) by the following equality:

$$
\frac{\pi(\boldsymbol{x}^{(k)} | \psi_k, \theta_{k-1})}{\pi(\boldsymbol{x}^{(k)} | \theta_{k-1})} = \frac{\pi(\psi_k | \boldsymbol{x}^{(k)}, \theta_{k-1})}{\pi(\psi_k | \theta_{k-1})} = \frac{\pi(\psi_k | \boldsymbol{x}^{(k)})}{\pi(\psi_k | \theta_{k-1})} = 1. \tag{A2}
$$

## B Proof of Theorem 3.1 and Theorem 3.2

For convenience, we denote the trajectory observation $(\boldsymbol{x}^{(k)}, \boldsymbol{R}^{(k)})$ as $\boldsymbol{z}_k$, and assume $\boldsymbol{z}_k \in \mathcal{Z}$ be a compact set. The Latent Trajectory Framework can be written in a general form as

$$
\begin{aligned}
\psi_k &= \psi_{k-1} + \epsilon_k \nabla_\psi \tilde{L}(\theta_{k-1}, \psi_{k-1}, \boldsymbol{z}_k) + \sqrt{2\epsilon_k} e_k, \\
\theta_k &= \theta_{k-1} + \upsilon_k \tilde{g}(\theta_{k-1}, \psi_k, \boldsymbol{z}_k),
\end{aligned} \tag{A3}
$$

where $\epsilon_k$ denotes the learning rate, $e_k$ is a standard Gaussian noise, $\nabla_\psi \tilde{L}(\theta_{k-1}, \psi_{k-1}, \boldsymbol{z}_k)$ denotes an unbiased estimate of $\nabla_\psi L(\theta_{k-1}, \psi_{k-1}) = \nabla_\psi \log \pi(\psi_{k-1} | \theta_{k-1})$, and $\tilde{g}(\theta_{k-1}, \psi_k, \boldsymbol{z}_k)$ is an unbiased estimator of $g^{ac}_{\psi_k}(\theta_{k-1})$. Convergence of adaptive stochastic gradient MCMC algorithms has been studied in Deng et al. (2019), Dong et al. (2023) and Liang et al. (2025). The convergence theory of LTF can be established by slightly modifying some of the assumptions used therein.

**Notation:** We use $\mathbb{E}_\psi[u(\theta, \psi)]$ to denote the expectation of $u(\theta, \psi)$ with respect to the conditional distribution $\pi(\psi | \theta)$, and use $\mathbb{E}[u(\cdot)]$ to denote the expectation with respect to the joint distribution of all the variables involved in the integrand $u(\cdot)$.

**Assumption A1** *The step size sequence $\{\upsilon_k\}_{k\in\mathbb{N}}$ is a positive decreasing sequence of real numbers such that*

$$
\lim_{k\to\infty} \upsilon_k = 0, \quad \sum_{k=1}^\infty \upsilon_k = \infty. \tag{A4}
$$

*There exist $\delta > 0$ and a stationary point $\theta^*$ such that for any $\theta \in \Theta$,*

$$
\langle \theta - \theta^*, g(\theta) \rangle \le -\delta \|\theta - \theta^*\|^2,
$$

where $g(\theta) = \mathbb{E}_\psi[g_\psi^{ac}(\theta)]$ and, in addition,

$$\liminf_{k\to\infty} 2\delta \frac{v_k}{v_{k+1}} + \frac{v_{k+1} - v_k}{v_{k+1}^2} > 0, \tag{A5}$$

where $\|\cdot\|$ denotes the $L^2$-norm.

**Assumption A2** $L(\theta, \psi)$ is $M$-smooth on $\theta$ and $\psi$ with $M > 0$, and $(m, b)$-dissipative on $\psi$ for some constants $m > 1$ and $b > 0$. In other words, for any $\psi, \psi', \psi'' \in \Psi$ and $\theta, \theta' \in \Theta$, the following inequalities are satisfied:

$$\|\nabla_\psi L(\theta, \psi') - \nabla_\psi L(\theta', \psi'')\| \le M\|\psi' - \psi''\| + M\|\theta - \theta'\|, \tag{A6}$$

$$\langle \nabla_\psi L(\theta^*, \psi), \psi \rangle \le b - m\|\psi\|^2, \tag{A7}$$

where $\theta^*$ is a stationary point as defined in Assumption A1.

Assumption A1 is a critical and standard assumption in the convergence of SGMCMC algorithms. In the context of deep neural networks, the dissipativity condition can be easily achieved by imposing a Gaussian prior on the critic network parameter, which further guarantees convergence.

**Lemma B.1** $\|\nabla_\psi L(\theta, \psi)\|^2 \le 3M^2\|\psi\|^2 + 3M^2\|\theta - \theta^*\|^2 + 3B^2$ for some constant $B$.

PROOF: Follow the proof of Lemma A.1 in Dong et al. (2023). □

**Assumption A3** Let $\zeta_k = \nabla_\psi \tilde{L}(\theta_k, \psi_k, z_k) - \nabla_\psi L(\theta_k, \psi_k)$. Assume that $\zeta_k$'s are mutually independent white noises, and they satisfy the conditions

$$\mathbb{E}(\zeta_k|\mathcal{F}_k) = 0, \quad \mathbb{E}\|\zeta_k\|^2 \le \delta_{\tilde{L}}(M^2\mathbb{E}\|\psi_k\|^2 + M^2\mathbb{E}\|\theta_k - \theta^*\|^2 + B^2), \tag{A8}$$

where $\delta_{\tilde{L}}$ and $B$ are positive constants, and $\mathcal{F}_k = \sigma\{\theta_1, \psi_1, \theta_2, \psi_2, \dots, \theta_k, \psi_k\}$ denotes a $\sigma$-filtration.

**Assumption A4** There exist positive constants $M$ and $B$ such that for all $z \in \mathcal{Z}$,

$$\|\tilde{g}(\theta, \psi, z)\| \le M^2\|\theta - \theta^*\|^2 + M^2\|\psi\|^2 + B^2,$$

where $\tilde{g}(\theta, \psi, z)$ is as defined in (A3).

By the formulation defined in section 3.2, let $g(\theta) = \mathbb{E}_{(\psi,z)}[\tilde{g}(\theta, \psi, z)|\theta]$ and $\eta = \tilde{g}(\theta, \psi, z) - g(\theta)$. Since $\mathbb{E}_{(\psi,z)}[\|\tilde{g}(\theta, \psi, z)\|^2|\theta] = \|g(\theta)\|^2 + \mathbb{E}_{(\psi,z)}[\|\eta\|^2|\theta]$, this implies $\mathbb{E}\|g(\theta)\|^2 \le \mathbb{E}\|\tilde{g}(\theta, \psi, z)\|^2$ and $\mathbb{E}\|\eta\|^2 \le \mathbb{E}\|\tilde{g}(\theta, \psi, z)\|^2$.

**Lemma B.2** (Uniform $L^2$ bounds) Suppose Assumptions A1-A4 hold. If the following conditions are satisfied:

$$\epsilon_k = \frac{C_\epsilon}{c_\epsilon + k^\alpha}, \quad v_k = \frac{C_v}{c_v + k^\beta}, \tag{A9}$$

for some constants $C_\epsilon > 0$, $c_\epsilon > 0$, $C_v > 0$, $c_v > 0$, $\alpha, \beta \in (0, 1]$, and $\beta \le \alpha \le \min\{1, 2\beta\}$. Then there exist constants $G_\psi$ and $G_\theta$ such that $\mathbb{E}\|\psi_k\|^2 \le G_\psi$ and $\mathbb{E}\|\theta_k - \theta^*\|^2 \le G_\theta$ for all $k = 0, 1, 2, \dots$.

PROOF: Follow the proof of Lemma A.2 in Dong et al. (2023). We slightly modify Assumption 4 in Dong et al. (2023) by Assumption A4, where the stochastic gradient is replaced with $\hat{g}_\psi^{ac}(\theta)$. Then the proof is straight forward.

□

**Assumption A5** (Solution of Poisson equation) For any $\theta \in \Theta$, $\psi \in \Psi$, and a function $\mathcal{V}(\psi) = 1 + \|\psi\|$, there exists a function $\mu_\theta$ on $\Psi$ that solves the Poisson equation $\mu_\theta(\psi) - \mathcal{T}_\theta\mu_\theta(\psi) = g_\psi^{ac}(\theta) - g(\theta)$, where $\mathcal{T}_\theta$ denotes a probability transition kernel with $\mathcal{T}_\theta\mu_\theta(\psi) = \int_\Psi \mu_\theta(\psi')\mathcal{T}_\theta(\psi, \psi')d\psi'$, such that

$$g_{\psi_{k+1}}^{ac}(\theta_k) = g(\theta_k) + \mu_{\theta_k}(\psi_{k+1}) - \mathcal{T}_{\theta_k}\mu_{\theta_k}(\psi_{k+1}), \quad k = 1, 2, \dots. \tag{A10}$$

Moreover, for all $\theta, \theta' \in \Theta$ and $\psi \in \Psi$, we have $\|\mu_\theta(\psi) - \mu_{\theta'}(\psi)\| + \|\mathcal{T}_\theta\mu_\theta(\psi) - \mathcal{T}_{\theta'}\mu_{\theta'}(\psi)\| \le \varsigma_1\|\theta - \theta'\|\mathcal{V}(\psi)$ and $\|\mu_\theta(\psi)\| + \|\mathcal{T}_\theta\mu_\theta(\psi)\| \le \varsigma_2\mathcal{V}(\psi)$ for some constants $\varsigma_1 > 0$ and $\varsigma_2 > 0$.

**Proof of Theorem 3.1**

PROOF: For Algorithm 1, we assume that the sample size of auxiliary $\tilde{\psi}$-samples is sufficiently large, ensuring the denominator estimator in Eq. (13) converges almost surely to its mean value (Teh et al., 2016). Therefore, the resulting stochastic gradient (14) is almost surely unbiased.

Dong et al. (2023) proved the result (16) for a more general adaptive Langevinized ensemble Kalman filter (LEnKF) algorithm, which is equivalent to an adaptive pre-conditioned SGLD algorithm. Extending their proof to Algorithm 1 is straight forward. $\square$

**Assumption A6** *The probability law $\mu_0$ of the initial hypothesis $\theta_0$ has a bounded and strictly positive density $p_0$ with respect to the Lebesgue measure on $\mathbb{R}_{d_\psi}$, and*

$$\kappa_0 := \log \int_{\mathbb{R}^{d_\psi}} e^{\|\theta\|^2} p_0(\theta)d\theta < \infty.$$

**Proof of Theorem 3.2**

PROOF: This theorem is proved in Liang et al. (2025) with the same Assumptions A1-A6. For Algorithm 1, we only need to assume that the sample size $m$ of auxiliary $\tilde{\psi}$-samples is sufficiently large, as explained in the proof of Theorem 3.1. $\square$

## C   Experiment Settings

### C.1   Escape environment

In this experiment, both $\pi_\theta$ and $V_\psi$ are approximated by deep neural networks with two hidden layers of sizes (128, 128). The agent updates the network parameters every 50 interactions, for a total of $10^6$ action steps. Each experiment is replicated for 100 times. For initial exploration, an entropy penalty coefficient of 0.01 is added, and gradually decay to 0. To achieve sparse deep neural network, we follow the suggestion in Sun et al. (2022) to impose mixture Gaussian prior onto both network parameters:

$$\theta, \psi \sim (1 - \lambda)\mathcal{N}(0, \sigma_0^2) + \lambda\mathcal{N}(0, \sigma_1^2) \tag{A11}$$

where $\lambda \in (0, 1)$ is the mixture proportion and $\sigma_0^2$ is usually set to a small number compare to $\sigma_1^2$. We set $\sigma_1 = 0.01$, $\sigma_0 = 0.001$ and $\lambda = 0.5$ in all LTF-enhanced algorithms. For indoor escape environment, the reward is given by $\mathcal{N}(-1, 0.01)$; that is, we set $\sigma^2 = 0.01$. To make the estimated return $y_t = R_t$ stationary, the reward at the goal state is set to $\mathcal{N}(-1, \frac{0.01}{1-\gamma^2})$, where the discount factor $\gamma = 0.9$. To guarantee the convergence of LTF, we set the decay policy learning rate as $\upsilon_k = O(\frac{1}{k^{0.5}})$ and constant critic learning rate $\epsilon_k = 2 \times 10^{-4}$. The sample size $L$ in (12) is set to 50, and the auxiliary sample size $m$ in (13) is set to 5.

In practical implementation, drawing samples from the conditional distribution $\pi_\mathcal{N}(\psi_k|\boldsymbol{x}_k)$ can be performed with a short sub-loop of SGMCMC updates, which we set the length to be 10. That is, for each iteration $k$, we repeat the $\psi$-sampling update 10 times. The sub-loop sampling scheme is given by

$$\psi_{k,\ell} = \psi_{k,\ell-1} + \frac{\epsilon_{k,\ell}}{2}\hat{w}_{k,\ell}\bigg\{ \sum_{i=1}^{n} \nabla_\psi \log \pi(R_i^{(k)}|x_i^{(k)}, \psi_{k,\ell-1}) + \frac{n}{\mathcal{N}}\nabla_\psi \log \pi(\psi_{k,\ell-1}) \bigg\} + e_{k,\ell} \tag{A12}$$

where the sub-loop is indexed by $\ell$. And the importance weight can be calculated by

$$\hat{w}_{k,\ell} = \frac{\pi(\boldsymbol{R}^{(k)}|\boldsymbol{x}^{(k)}, \psi_{k,\ell-1})}{\frac{1}{m+1}\sum_{j=1}^{m} \pi(\boldsymbol{R}^{(k)}|\boldsymbol{x}^{(k)}, \tilde{\psi}_j) + \frac{1}{m+1}\pi(\boldsymbol{R}^{(k)}|\boldsymbol{x}^{(k)}, \psi_{k,\ell-1})}.$$

where $m$ denote the number of auxiliary samples and the importance weight is bounded by $m + 1$. The boundedness of the importance weights $\hat{w}_{k,\ell}$'s further ensures the stability of SGMCMC sampling step. We note that including the $\psi_{k,l-1}$-term in the denominator is reasonable. As implied by the definition of

the importance weight $w_k = \frac{\pi(\boldsymbol{R}^{(k)}|\boldsymbol{x}^{(k)},\psi_k)}{\pi(\boldsymbol{R}^{(k)}|\boldsymbol{x}^{(k)})}$, the numerator term should be part of the denominator and, therefore, we need to include $\psi_{k,l-1}$ as an auxiliary sample of $\tilde{\psi}$. Furthermore, we refer to Theorem 1 of Song et al. (2020) for the sample equally weighted formula in calculating the denominator.

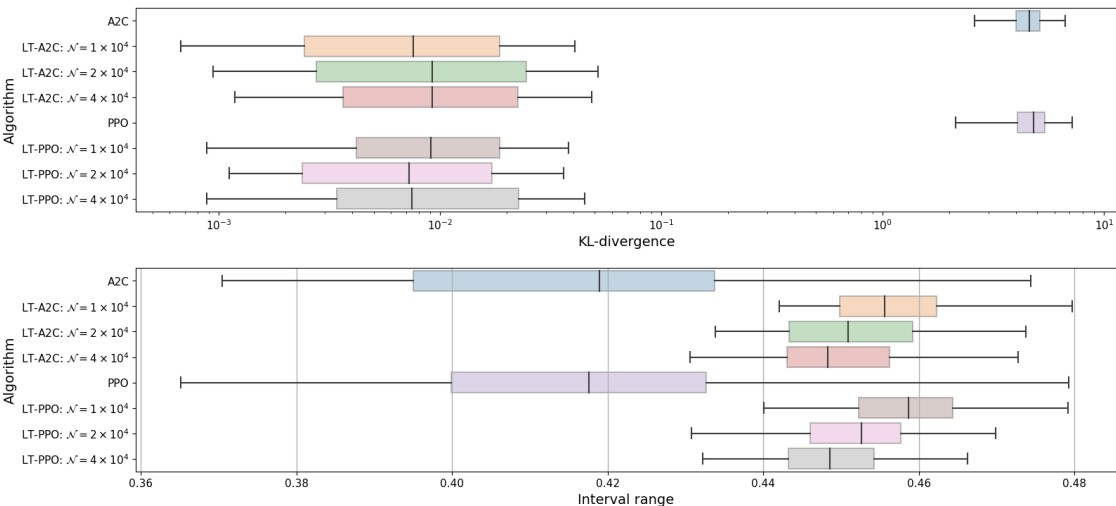

Figure A1: KL-divergence (top plot) and interval widths (bottom plot) achieved by A2C, PPO, LT-A2C, and LT-PPO for the Indoor Escape example. The results were summarized from 100 independent runs for each method.

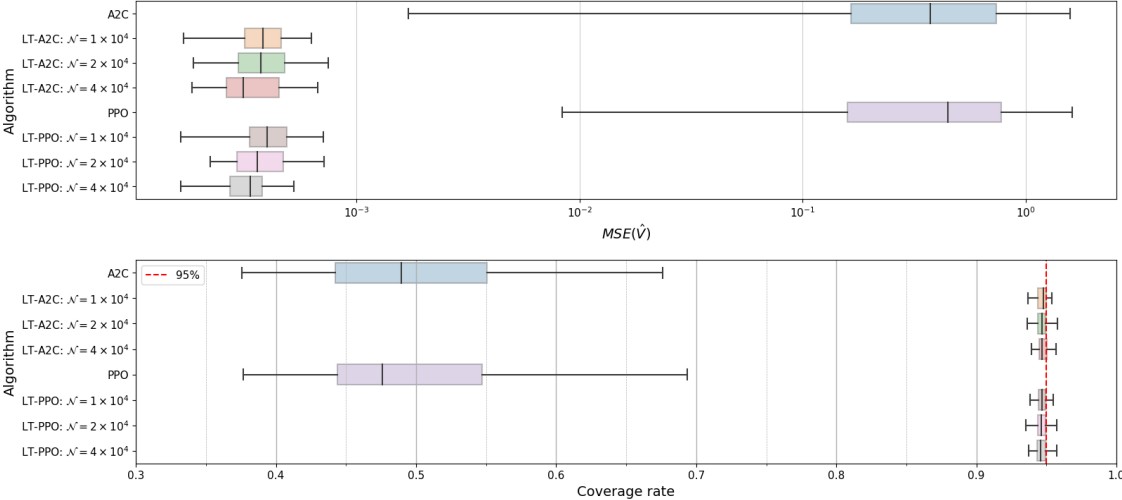

Figure A2: *Top plot*: MSE of $\hat{V}$; *Bottom plot*: Coverage rate of the 95% prediction interval for the value $V^*(s)$. The results were computed from 100 independent runs for each method.

## C.2 PyBullet environment

In this experiment, we conduct experiments on PyBullet environments, including Ant, HalfCheetah, Hopper, Reacher, and Walker2D. The training framework and hyperparameters of A2C and PPO are based on RL baselines3 zoo, and our LT-A2C and LT-PPO is implemented on top of Stable-Baselines3 Raffin et al. (2021). The hyperparameters are listed in Tables A1, A2, A3, and A4. Any hyperparameter not specified in these tables is set to the default value used in its corresponding vanilla counterpart. The actor and critic networks in LT-A2C and LT-PPO follow the default configurations of A2C and PPO, respectively, as implemented in

the Stable-Baselines3 Zoo. There are 2 types of learning rate, constant and linear decay. To balance between exploration and exploitation in LT-enhanced algorithms, we adopt an annealing technique, where the pseudo population size increases as training steps increase, starting from 500. This method allows the algorithm to gradually shift from exploration to exploitation, improving overall performance and stability. A2C optimizes both the actor and critic networks using RMSprop, whereas PPO employs the Adam optimizer for both networks. In contrast, LT-A2C and LT-PPO update the actor network using RMSprop while performing SGHMC sampling for the critic parameters. For the prior distribution, both LT-A2C and LT-PPO adopt the same Gaussian mixture prior used in the Escape environment.

In theory, the auxiliary sampling step requires large sample size to guarantee a good approximation. To improve the sampling efficiency of the auxiliary sampling step, we modify the approximation procedure of the importance weight. We replace the auxiliary samples $\tilde{\psi}_j$'s with the SGMCMC samples $\psi_{k,\ell}$ derived in (A12). The importance weight can then be approximated by

$$\hat{w}_{k,\tilde{\ell}} = \frac{\pi(\boldsymbol{R}^{(k)}|\boldsymbol{x}^{(k)}, \psi_{k,\tilde{\ell}-1})}{\frac{1}{\tilde{\ell}} \sum_{\ell=0}^{\tilde{\ell}-1} \pi(\boldsymbol{R}^{(k)}|\boldsymbol{x}^{(k)}, \psi_{k,\ell})}.$$

With this modification, we can eliminate the auxiliary sampling step and further lower the computation complexity and memory complexity. The rationale behind this replacement is that the drift term $\nabla_{\tilde{\psi}} \log \pi(\tilde{\psi} \mid \boldsymbol{x}^{(k)}, \boldsymbol{R}^{(k)})$, used in (A12), provides an unbiased estimator of the drift $\nabla_{\tilde{\psi}} \log \pi(\tilde{\psi} \mid \boldsymbol{x}^{(k)})$ that governs the dynamics in (12) for simulations of auxiliary samples.

Table A1: Hyperparameters for A2C and LT-A2C

| Environment | HalfCheetah | | Hopper | |
|---|---|---|---|---|
| Hyperparameters | LT-A2C | A2C | LT-A2C | A2C |
| learning rate | lin 0.00067 | lin 0.00096 | lin 0.00042 | lin 0.00096 |
| $\sigma$ (observation) | 0.1 | - | 0.1 | - |
| $\mathcal{N}$ | 50000 | - | 10000 | - |
| $\gamma$(discount factor) | 0.95 | 0.99 | 0.99 | 0.99 |
| gae-$\lambda$ | 0.9 | 0.9 | 1.0 | 0.9 |
| train batch | 32 | 32 | 32 | 32 |
| training steps | 2e6 | 2e6 | 2e6 | 2e6 |

Table A2: Hyperparameters for A2C and LT-A2C (cont.)

| Environment | Reacher | | Walker2D | |
|---|---|---|---|---|
| Hyperparameters | LT-A2C | A2C | LT-A2C | A2C |
| learning rate | lin 0.00096 | lin 0.0008 | lin 0.00037 | lin 0.00096 |
| $\sigma$ (observation) | 0.1 | - | 0.1 | - |
| $\mathcal{N}$ | 1000 | - | 500 | - |
| $\gamma$(discount factor) | 0.99 | 0.99 | 0.99 | 0.99 |
| gae-$\lambda$ | 1.0 | 0.9 | 1.0 | 0.9 |
| train batch | 32 | 32 | 32 | 32 |
| training steps | 2e6 | 2e6 | 2e6 | 2e6 |

Table A3: Hyperparameters for PPO and LT-PPO

| Environment | HalfCheetah | | Hopper | |
|---|---|---|---|---|
| Hyperparameters | LT-PPO | PPO | LT-PPO | PPO |
| learning rate | 3e-5 | 3e-5 | 3e-5 | 3e-5 |
| $\sigma$ (observation) | 0.1 | - | 0.1 | - |
| $\mathcal{N}$ | 50000 | - | 50000 | - |
| $\gamma$(discount factor) | 0.99 | 0.99 | 0.99 | 0.99 |
| gae-$\lambda$ | 0.9 | 0.9 | 0.9 | 0.9 |
| train batch | 128 | 128 | 128 | 128 |
| training steps | 2e6 | 2e6 | 2e6 | 2e6 |

Table A4: Hyperparameters for PPO and LT-PPO (cont.)

| Environment | Reacher | | Walker2D | |
|---|---|---|---|---|
| Hyperparameters | LT-PPO | PPO | LT-PPO | PPO |
| learning rate | 3e-5 | 3e-5 | 3e-5 | 3e-5 |
| $\sigma$ (observation) | 0.1 | - | 0.1 | - |
| $\mathcal{N}$ | 50000 | - | 50000 | - |
| $\gamma$(discount factor) | 0.99 | 0.99 | 0.99 | 0.99 |
| gae-$\lambda$ | 0.9 | 0.9 | 0.9 | 0.9 |
| train batch | 64 | 64 | 128 | 128 |
| training steps | 2e6 | 2e6 | 2e6 | 2e6 |

