# OpenReview forum: "Latent Trajectory: A New Framework for Deep Actor-Critic Reinforcement Learning with Uncertainty Quantification"
_TMLR — Accepted by TMLR_

### Review · Reviewer_udfF · 2025-07-05

**Summary Of Contributions:**

This paper proposes a framework that treats transition trajectories as latent variables, which then motivates the adaptive Stochastic Gradient Markov Chain Monte Carlo algorithm to train deep actor-critic models, which naturally accounts for the interdependence between the actor and critic updates.

**Audience:**

Yes

**Broader Impact Concerns:**

None.

**Claims And Evidence:**

Yes

**Requested Changes:**

I would suggest the authors to motivate the problem studied in this paper in a better and more convincing way. For the empirical results, the authors should either use the same hyperparameter for both A2C and LTF-A2C, or conduct an ablation study to show that the performance improvement is due to the LTF-enhanced approach, not hyperparameter tuning.

**Strengths And Weaknesses:**

**Strengths**

Overall a well-organized, mathematically clear paper that is easy to follow.

**Weaknesses**

* The motivation of incorporating uncertainty quantification (UQ) into RL needs further elaboration, especially how it impacts current methods. As shown in Figure 7, there is very little difference between the performance of the original PPO algorithm and the LTF-enhanced PPO (LT-PPO) approach, suggesting that it might not be a serious issue in deep RL.

* In Table A1 and A2, they show that LT-A2C and A2C use different hyperparameters (e.g., GAE factor $\lambda$ and discount factor $\gamma$). An ablation study is needed to examine their effects.

---

> ### Author Response · Authors · 2025-08-09
>
> **[W1] The motivation of incorporating uncertainty quantification (UQ) into RL needs further elaboration, especially how it impacts current methods. As shown in Figure 7, there is very little difference between the performance of the original PPO algorithm and the LTF-enhanced PPO (LT-PPO) approach, suggesting that it might not be a serious issue in deep RL.**
>
> We appreciate the reviewer’s question regarding the motivation for incorporating uncertainty quantification (UQ) into reinforcement learning (RL). While conventional RL algorithms like PPO and A2C aim to maximize expected return, they do not explicitly estimate the confidence in their value predictions. UQ is particularly important in scenarios where safety, robustness, and efficient exploration are critical. Accurate UQ allows an agent to: (i) enhance deep exploration and thus  optimal policy search; as shown in Figure 1* (https://ibb.co/hJ3k8ZYB) and Figure 2*(https://ibb.co/9xyyN0B), the proposed method does improve the reward in certain benchmark environments, while making the results comparable in others, and (ii) provide calibrated confidence intervals for policy evaluation.
>
> **[W2]  In Table A1 and A2, they show that LT-A2C and A2C use different hyperparameters (e.g., GAE factor  and discount factor ). An ablation study is needed to examine their effects.**
>
> Now we use the same $\gamma$ and $ \mathrm{GAE}-\lambda$ for both A2C and LT-A2C in each environment. From Figure 2*, it clearly shows that the proposed method outperforms existing methods in some environments, while performing comparably in others.  Additionally, our method is able to quantify the uncertainty of the resulting values at each time step.

---

### Review · Reviewer_Dfgk · 2025-07-06

**Summary Of Contributions:**

The authors introduce a method for integrating uncertainty estimation of the value function into actor critic algorithms. By sampling the value function weights and weighting the update of the value function from these samples, this framework is able to incorporate actor informed uncertainty estimates into the training of the critic. This algorithm is shown to have certain theoretical convergences and is verified on a toy environment and a selection of pybullet environments.

**Audience:**

Yes

**Claims And Evidence:**

No

**Requested Changes:**

The key changes (outside of the minor citation/figure stuff) are:
- Consolidation of grid results
- Ensuring the necessary algorithmic adjustments are in the main text that are required for scaling
- Further consideration of time/compute

Which are outlined in greater detail above.

**Strengths And Weaknesses:**

Pros:
- The core idea is nicely built upon previous work and advances in an interesting and underexplored area of uncertainty based RL methods
- Decoupling the actor/critic by treating the trajectories as the latent variables is a neat idea and this work could lead to interesting follow ups that bridge it with Bayesian approaches
- The flexibility to integrate the framework into a variety of actor critic methods increases its useability
- The toy model evaluation helps showcase how and why the framework provides an advantage

Cons:
- Figure 7 could be improved. Since each variant is connected/looked at with its LT pair, it’s worth visually connecting them in some way (e.g. make them the same color and vary the line style or something). The dpi should be higher (it looks a bit blurry if you zoom in). All three columns are not necessary, pick the column to use, then make the 4 environments in a row and move the rest to the appendix.
- Too much time is spent on the toy model results. They are interesting and quite helpful to an extent, but 5 figures and 2 tables is overkill. Consolidate them to focus on the key points of explaining how and why the framework is advantageous, and move the rest to the appendix. Table 2 is important and very good to have and should center more in the explanations. The figures should highlight the answer to the question “why is it that existing algorithms fail to capture this in the same way?”
- “Regarding computational complexity, we note that although LTF requires additional SGMCMC sampling for the critic network at each iteration, the per-iteration complexity of SGLD and SGHMC is comparable to that of SGD. As a result, the overall time complexity remains on par with standard iterative optimization methods, supporting the scalability of the proposed method. The LTF methods can be readily applied to large-scale neural networks.” This is a top question I had while reading the paper. While the per iteration complexity for many SGMCMC methods is comparable to SGD, the real question is how many iterations are necessary. To this end, the work (possibly in the appendix) would benefit from a wall clock/GPU hours plot comparison.
- Related to the previous point, one key thing is how big of `m` is necessary to get good results. For the scaled up pybullet work, the appendix showed “With this modification, we can eliminate the auxiliary sampling step and further lower the computation complexity and memory complexity.” This seems like a pretty important aspect that is relegated to the appendix?
- “In this experiment, we omit the reports for the uncertainty of the value functions, as their true values are unknown.” An interesting plot (although not one useful for quality comparison, but one that could be an interesting insight into the functioning of the algorithms), is to plot the uncertainty estimate for a variety of algorithms over training and compare what they look like (there isn’t a “right” or “wrong”, but might elucidate where some things happen).
- There could be some additional comparison/literature included, some works such as https://arxiv.org/abs/1711.10789, https://arxiv.org/abs/1905.09638, https://arxiv.org/abs/2103.00445, https://arxiv.org/abs/2201.01666, https://proceedings.mlr.press/v78/kalweit17a/kalweit17a.pdf, etc.
- The quality/importance of the SGMCMC algorithm is not discussed or ablated. It would be interesting to see how crucial the success of that algorithm is to the success of the outer loop.

---

> ### Author Response · Authors · 2025-08-09
>
> **[W1] Figure 7 could be improved. ...**
>
> We have revised Figure 7, see Figure 2*(https://ibb.co/9xyyN0B) of this reply.  Each curve in the plots represents the mean of the best-so-far reward obtained by each method in 100 independent runs. The shaded region indicates the 95\% confidence interval, computed as the mean ± 1.96 times the standard error. It clearly shows that the proposed method outperforms existing methods in some cases, while performing comparably in others. Additionally, our method is able to quantify the uncertainty of the resulting values at each time step.
>
> **[W2] Too much time is spent on the toy model results....**
>
> The toy model example has been revised according to your suggestions. Specifically, we moved Figure 4 and Figure 5 to the Appendix, while adding a subplot (see Figure 1*(a) (https://ibb.co/hJ3k8ZYB) of this reply) to indicate that the proposed method outperforms A2C in optimal policy search.
>
> Regarding Table 2, we note that our experiments highlight the inherent difficulty of the problem we are addressing: when neural networks are involved, the existing methods can be challenging to calibrate, often leading to inaccurate coverage rates.
>
> **[W3] Regarding computational complexity, ...**
>
> As suggested, we have compared the computational complexity of the different methods in terms of wall-clock time; see Figure 3* (https://ibb.co/S71R3kjj). It presents reward–wall time plots for A2C and LT-A2C with different numbers of SGMCMC steps ($5$, $10$, and $15$). All experiments use the same number of actor network updates. The results show that LT-A2C performance improves as the number of SGMCMC steps increases, albeit at the cost of longer computation time.
>
> **[W4] Related to the previous point, one key thing is how big of $m$ is necessary to get good results.**
>
> In the revision, we have experimented with different values of $m$: 5, 10, and 15. The results in Figure 3* suggest that $m=10$ provides a good balance, although larger values of $m$ tend to yield higher rewards.
>
> **[W5] For the scaled up pybullet work, the appendix showed “With this modification, we can eliminate the auxiliary sampling step and further lower the computation complexity and memory complexity.” This seems like a pretty important aspect that is relegated to the appendix?**
>
> The rationale behind this replacement is that the drift term $ \nabla_{\tilde{\psi}} \log \pi(\tilde{\psi} \mid x^{(k)}, R^{(k)}) $, used in (A12), provides an unbiased estimator of the drift $\nabla_{\tilde{\psi}} \log \pi(\tilde{\psi} \mid x^{(k)}) $ that governs the dynamics in (12) for simulations of auxiliary samples. It corresponds to the case $L=1$ in (12).
>
> **[W6] “In this experiment, we omit the reports for the uncertainty of the value functions, as their true values are unknown.” An interesting plot (although not one useful for quality comparison, but one that could be an interesting insight into the functioning of the algorithms), is to plot the uncertainty estimate for a variety of algorithms over training and compare what they look like (there isn’t a “right” or “wrong”, but might elucidate where some things happen).**
>
> Since our experiments are implemented in the Stable-Baselines3 Zoo, extracting per-iteration samples from the training loop is nontrivial. To address the reviewer’s concern, we instead demonstrate convergence in the Escape environment. As shown in Figure 4* (https://ibb.co/RpzyYfnS), the empirical mean of LT-A2C’s state-value samples matches the true state value, whereas A2C’s samples exhibit bias. This plot should resemble the performance of the two methods in other environments.
>
> **[W7] There could be some additional comparison...**
>
> These works will be cited in the revision. Thank you for pointing them out to us.
>
> **[W8] The quality/importance of the SGMCMC algorithm is not discussed or ablated. It would be interesting to see how crucial the success of that algorithm is to the success of the outer loop.**
>
> Thank you for this thoughtful question. In the revision, we experimented with different values of
> $m=5$, $10$, $15$ (see Figure 3* of the reply), which partially addresses this question. While a high-quality SGMCMC algorithm can enhance the performance of the proposed method, it is not critically essential.

---

> > ### Comment · Reviewer_Dfgk · 2025-08-25
> >
> > I appreciate the authors response, the new figures are quite nice and will make the paper better. In the original review I had three main points for improvement (Consolidation of grid results, Ensuring the necessary algorithmic adjustments are in the main text that are required for scaling, Further consideration of time/compute). These figures help meet the third requirement and first requirement (by shrinking the total space spent on the grid). Based on what the authors have said, I believe the text will also be improved to address the second point. I consider my list of primary feedback for improvement to be completed.

---

### Review · Reviewer_eHaw · 2025-07-25

**Summary Of Contributions:**

This paper focuses on how we can successfully quantify the uncertainty in our Q and value-function estimates in deep RL actor-critic algorithms. Specifically, prior works in uncertainty quantification in RL and deep RL (without actor-critic) have shown its importance. The idea of this paper is to introduce a novel framework that treats the transition trajectories as latent variables, and then use it to improve any existing actor-critic deep RL algorithm. The authors provide theoretical convergence results for the proposed method, and then conduct numerical simulations for validating it.

**Audience:**

No

**Broader Impact Concerns:**

/

**Claims And Evidence:**

No

**Requested Changes:**

- The presentation of sections 2 and 3 should be changed almost completely.
- Additional experiments that clarify precisely that the proposed method improves over the sota should be conducted.
- Fix the other issues I identified in the block above.

**Strengths And Weaknesses:**

# Strengths

- Deep actor-critic algorithms are important for the RL community.
- Both theoretical and empirical validation of the proposed method.

# Weaknesses

The paper suffers from three main limitations.

## 1) The empirical validation is not significant.
- Regarding the escape environment, is not clear why we should care about, e.g., the notion of policy diversity (which is not a meaningful objective in general, and the authors do not provide enough arguments in support of using it), and not about the expected return induced by the learned policy. This is seen also in Table 1, where it is not shown the difference in performance (i.e., expected return) between the policies learned through PPO/A2C and the proposed method.
- Regarding the experiments in the PyBullet environment, that compare the expected returns of the policies learned by the various algorithms, it is clear from FIg. 7 that the std is quite large, so we cannot confidently conclude that the proposed method actually improves over the existing ones.
## 2) The presentation is quite confusing.
Especially in Sections 2 and 3, where the technical contribution of the paper is presented, more clarity and intuition would be appreciated. To make some examples, in Section 2:
- Why is $R_t$ "denoted by $Q^{\pi_\theta}$"? What does it mean?
- What is the stationary distribution $\pi(x|\theta_{k-1})$? I don't get it.
- What is an optimal policy? It is not defined.
- Why do you include also $R^{(k)}$ in the transition trajectory?
- What is a pseudo-population?
- Why you say that the training is completed when Eqs. (3)-(5) are obtained? I mean, the goal is to maximize the expected return, not finding a point where the gradient is 0.

## 3) Other parts of the submission lack quality.
- In the discussion of related works, some are missing. E.g., "Kalman meets Bellman - Improving Policy Evaluation through Value Tracking".
- Why do you claim that distributional RL is about uncertainty quantification? It is not. In distributional RL, you care about the distribution of the return as a random variable. Instead, in previous works on uncertainty quantification (e.g., "Kalman Temporal Differences"), the focus is on the distance between your estimate of the expected return, and the true expected return. These are two quite different objects.

# Questions

In section 4.1, what is the coverage rate?

---

> ### Author Response · Authors · 2025-08-09
>
> ### [W1] The empirical validation is not significant.
> **Regarding the escape environment, ......**
>
> In the revision, we will add Figure 1*(https://ibb.co/hJ3k8ZYB) to show that the proposed method substantially improves reward over the baseline A2C method, while Table 1 highlights its uncertainty-quantification capability.
> Specifically, Figure 1*(a) plots the best-so-far reward versus time steps for each method: the proposed method reaches the global optimum ($-18$) in every run, whereas A2C fails to do so in a substantial fraction of runs. Figure 1*(b) reports the success rates of the two methods in achieving the global optimum. The comparison suggests that the proposed method substantially outperforms A2C in identifying the optimal policy, while accurately quantifying the uncertainty of the resulting values.
>
> **Regarding the PyBullet environment, ......**
>
> In the revised version, we replot Figure 7 to improve clarity, see Figure 2*(https://ibb.co/9xyyN0B). Each curve represents the mean reward of the best model obtained up to the current iteration. The shaded region indicates the 95\% confidence interval, computed as the mean ± 1.96 times the standard error.
> It clearly shows that the proposed method outperforms existing methods in some cases, while performing comparably in others. Additionally, our method is able to quantify the uncertainty of the resulting values at each time step.
>
> Specifically, Figure 2*  (of this reply) shows that the proposed method significantly outperforms A2C and PPO in the Walker 2D example.
>
> ### [W2] The presentation is quite confusing. Especially in Sections 2 and 3, where the technical contribution of the paper is presented, more clarity and intuition would be appreciated. To make some examples, in Section 2:
>
> **Why is $R_t$ "denoted by $Q^{\pi_\theta}$"? What does it mean?**
> Sorry for the confusion. We are referring to the Q-value denoted by $Q^{\pi_\theta}$. The return $R_t$ serves as a Monte Carlo estimate of this Q-value.
>
> **What is the stationary distribution $\pi(x|\theta_{k-1})$? I don't get it.**
> Given an actor network parameter $\theta_{k-1}$, the policy $\pi_{\theta_{k-1}}$ is determined.
>             Based on this policy and the environment dynamics, we can define the stationary distribution
>             of the state-action transition trajectory $x = (s_0, a_0, s_1, a_1, \dots)$ as $\pi( x |\theta_{k-1})$. This has been clarified in the revision.
>
> **What is an optimal policy? It is not defined.**
> Optimal policy is the policy that maximizes the cumulative reward. In policy gradient method, maximizing the objective function is equivalent to solve the policy gradient equals to zero. This has been defined in the revision.
>
> **Why do you include also $R^{(k)}$ in the transition trajectory?**
> $R^{(k)}$ is the return generated through the interaction between the agent and the environment,
> which contains information about the policy. The latent trajectory framework treats these generated data as latent variables.
>
> **What is a pseudo-population?**
>
> Given a policy $\pi_\theta$, if we have infinitely many transition tuples, the value function can be exactly determined as $V^{\pi_\theta}$. In this case, the parameter $\psi$ of the value function approximator $V_\psi$ would collapse to a point estimate. To prevent the conditional distribution $\pi(\psi | \theta)$ from degenerating in this way, we define a pseudo-population that ensures this conditional distribution remains independent of the sample size.
>         This formulation enhances the robustness and flexibility of the training process with respect to batch size and facilitates the search for the optimal policy.
>
> **Why you say that the training is completed when Eqs. (3)-(5) are obtained? I mean, the goal is to maximize the expected return, not finding a point where the gradient is 0.**
>
> Maximizing the expected return $J(\theta)$ is through finding the critical point where $\nabla_\theta J(\theta) = 0$. Similarly for actor-critic and latent trajectory algorithms.
>
> ### [W3] Other parts of the submission lack quality.
>
> **In the discussion of related works, some are missing. E.g., "Kalman meets Bellman - Improving Policy Evaluation through Value Tracking"**
>
> In our revision, we will include related work on uncertainty quantification for single neural networks to improve clarity and make the submission easier to follow.
>
> **Why do you claim that distributional RL is about uncertainty quantification?**
>
> We appreciate your comment regarding the distinction between distributional RL and uncertainty quantification. You are correct that the primary goal of distributional RL is to model the full return distribution $Z^{\pi}(s,a)$, which captures the inherent randomness of the environment (aleatoric uncertainty), rather than to explicitly quantify epistemic uncertainty in the estimate of the expected return $Q^{\pi}(s,a)$. In the revision, we will remove it from comparisons.

---

### Decision · Action_Editor_xFAK · 2025-08-29

**Recommendation:** Accept with minor revision

**Additional Comments:**

The paper introduces a novel framework for deep actor-critic reinforcement learning that treats transition trajectories as latent variables, enabling the quantification of uncertainty via adaptive SGMCMC. Theoretical contributions are solid, providing convergence guarantees and a principled treatment of actor–critic interdependence. Revisions have clarified technical details, improved figures, and addressed most concerns from reviewers regarding presentation and algorithmic explanation.

Empirical results are convincing in specific tasks (e.g., Escape and Walker2D) but show modest or inconsistent gains in other benchmarks. While practical significance is somewhat limited, the framework’s originality and theoretical rigor make the work relevant to the TMLR audience. Minor revisions should focus on improving clarity in Sections 2–3 and framing empirical results to convey when and why the method outperforms baselines clearly.

**Audience:**

Yes

**Audience Explanation:**

The TMLR audience includes researchers in reinforcement learning and uncertainty quantification. While the practical impact is limited by mixed empirical results, the latent trajectory framework and theoretical treatment of uncertainty in actor–critic RL would be of interest to at least part of the readership.

**Claims And Evidence:**

Yes

**Claims Explanation:**

The theoretical claims are well supported and convincing. However, the empirical evidence is mixed: improvements are clear in some environments but inconsistent overall, scalability remains uncertain, and the presentation reduces clarity. Thus, while some claims are supported, others are not yet convincingly demonstrated.